# Biofertilizers from wastewater treatment as a potential source of mineral nutrients for growth of amaranth plants

Elisa Teófilo Ferreira[1], Sarah Corrêa Barrochelo[1], Sarah de Paula de Melo[1], Thainá Araujo[1], Augusto Cesar Coelho Xavier[1], Inês Cechin[1]*, Gustavo Henrique Ribeiro da Silva[2]

1 Department of Biological Sciences, Faculty of Sciences, UNESP – São Paulo State University, Bauru, Brazil, 2 Department of Civil Engineering, Faculty of Engeneering, UNESP – São Paulo State University, Bauru, Brazil

* ines.cechin@unesp.br

**Data Availability Statement:** All the data related to this study will be available in the following URL address https://doi.org/10.5061/dryad.kprr4xhb5.

## Abstract

Exploring alternative fertilizers is crucial in agriculture due to the cost and environmental impact of inorganic options. This study investigated the potential of sewage-derived biofertilizers on the growth and physiology of *Amaranthus cruentus* plants. Various treatments were compared, including control treatments with inorganic fertilizer and treatments with biofertilizers composed of microalgae, biosolids and reclaimed water. The following traits were investigated: photosynthetic pigments, gas exchange, growth, and leaf nutrient concentrations. The results showed that the concentrations of N, P, Cu, Fe Zn and Na nutrients, in the dry microalgae and biosolids, were quite high for the needs of the plants. The wet microalgae presented high concentration of Cu, Fe and Zn nutrients while reclaimed water contained high concentration of N, K, Ca and S. Na and Zn nutrients increased in the leaf of plants treated with dry microalgae and biosolid, respectively. At the beginning of the flowering phase, total chlorophyll and carotenoids contents were lower for plants grown with wet microalgae while for plants grown with higher doses of biosolid or reclaimed water total chlorophyll was increased, and carotenoids were not affected. Lower photosynthetic pigments under wet microalgae resulted in lower photosynthetic rates. On the other hand, amendments with dry microalgae and biosolid increased photosynthetic rates with the biosolid being the most effective. Higher applications of biosolid, wet and dry microalgae produced a considerable increase in shoot biomass of amaranth, with the dry microalgae being the most effective. Additionally, reclaimed water obtained after tertiary treatment of sewage with microalgae and biosolids applied alone showed promising effects on plant growth. Overall, these findings suggest that organic fertilizers derived from sewage treatment have the potential to enhance plant growth and contribute to sustainable agricultural practices.

## Introduction

As the global population continues to grow, so does the generation of waste and demand for resources. Unfortunately, the linear production model in which we are immersed often leads

**Funding:** The project was funded by the São Paulo Research Foundation, FAPESP (Grant numbers: 18/18367-1 [GHRS]; 2020/06459-9 [ETF] and 2020/10764-1 [SCB]; https://fapesp.br), the National Council for Scientific and Technological Development (Grant numbers: 308663/2021-7 [GHRS]; 309064/2018-0 [GHRS]; 427936/2018-7 [GHRS] and 334 [SPM]; www.cnpq.br). The funders had no role in study design, data collection and analysis, decision to publish, or preparation of the manuscript.

**Competing interests:** The authors have declared that no competing interests exist.

to inefficient use of resources and the generation of waste [1]. The circular economy proposes a shift in production chains towards closed-loop processes to improve resource efficiency and reduce waste [2, 3]. In this context, sewage is no longer considered as a waste, but as a valuable source for nutrient recovery [4, 5], which can be used to restore soil fertility and organic matter. Soil fertilization and the replacement of organic matter is essential due to the depletion of nutrients from the soil over the consecutive seasons, without which it is impossible to meet the food demand for a growing population [6]. As a result, exploring alternative sources of fertilizers is of great importance in agriculture due to the high cost and negative environmental impact of inorganic fertilizers. According to the literature, the resources recovered from the treatment of sanitary sewage could potentially meet up to 30% of the agricultural demand for nitrogen (N) [7] and 22% of the demand for phosphorus (P) [8, 9], thereby reducing the need for chemical fertilizers.

Sewage sludge is a residue rich in organic matter generated during the treatment of sewage in sewage treatment plants (STP). The characteristic of biological sewage sludge depends on the source of sewage which may contain heavy metals in addition to mineral nutrients and organic matter [10]. However, when subjected to additional treatments in order to reduce the water content and/or to stabilize the organic matter and remove pathogens [11, and references therein], it results in a biosolid that is easy to handle for disposal or use as a biofertilizer in agriculture because it has a large amount of nutrients and organic matter [12], and additionally, these drying and disinfection processes guarantee the suitability of the biosolids to safety regulations [13]. Several studies have been conducted in order to verify the benefits of biosolid application on several crops' growth and biochemical parameters [14–16]. In addition, the application of biosolid in areas undergoing reclamation can effectively improve the performance of introduced vegetation and the tolerance to potentially toxic elements [17] thus serving as a key tool for farm utilization [18]. The conventional sewage treatment processes have proven effective in managing sewage sludge but often present issues such as high energy consumption and the generation of residual sludge with environmental concerns. There is a growing interest in exploring alternative, sustainable sewage treatment options. The algal-based sewage treatment methods aims not only efficiently treat sewage but also harness the benefits thus contributing to the development of more sustainable and ecologically friendly solutions for sewage management.

Microalgae-based treatment of sewage is a promising method for nutrient recovery and wastewater treatment. After anaerobic treatment, the remaining effluent is favorable for the cultivation of microalgae as it retains most of the nutrients [19–21]. The ability of microalgae to use inorganic compounds such as N and P for their growth makes them efficient at removing nutrients from wastewater, which can reach up to 90% [22, 23]. Additionally, microalgae can help in disinfection by varying the pH and improving the quality of the final effluent [20, 24, 25]. The recovery of chemical energy and nutrients from microalgae biomass is also possible. According to [26], 68% of the chemical energy of microalgal biomass can be recovered, in the form of bio-oil (44%) and biogas (23%), in addition to the recovery of 44% of the N content and 91% of the P content of biomass in the form of fertilizer products. The organic matter of microalgae can also be used as a nutritive material with advantageous characteristics for crops and soil. From the organic matter of these microalgae, nutritive material can be obtained with advantageous characteristics for the crops and the soil, depending on the microalgae's ability to improve soil quality, its microflora and increase productivity [9, 27]. Studies have shown that organic microalgae fertilizer grown in wastewater can improve soil quality and increase crop productivity [5, 27, 28].

The reuse of treated domestic sewage in agriculture can provide multiple benefits. Reclaimed water is a valuable alternative to traditional irrigation water sources and can

contribute to reducing the pressure on freshwater resources [29]. The tertiary treatment of sewage with microalgae does not remove 100% of nutrients of the reclaimed water. Therefore, this treated sewage constitutes another by-product that can be destined for agriculture. Fertigation of plants with reclaimed water has proved to be very advantageous in regions where water availability is low [30, 31]. In addition to its value as a source of water, reclaimed water has also been shown to be efficient in increasing the productivity of several crops of economic interest due to the presence of essential nutrients necessary for plant growth despite the presence of some metals [32–34]. In Brazil, reclaimed water must meet the standards established in CONAMA resolution nº 357. Specific requirements are determined based on the type of effluent and the intended crop for which the reused water will be used [35]. According to the safety guidelines of the World Health Organization, the established parameter concerns the maximum permitted value of *E. coli*, which must be between $10^3$ and $10^4$, depending on the culture, whether it is consumed raw and whether there is direct contact with the edible part with the soil [36]

Although the use of biofertilizers available after sewage treatment still a controversial topic, sewage sludge, reclaimed water and microalgae are already being used in agriculture to promote the growth of several crops both in the greenhouse and in the field [34, 37–39]. The most common studied plants were cereals such as wheat and rice that are $C_3$ plants, therefore there is a great need to investigate other crops such as $C_4$ plants which have different nutritional requirements. Amaranth is a $C_4$ plant of great agronomic importance because of its high nutritional and nutraceutical values which provide health benefits. Manyelo *et al.* [40] reported higher calcium (Ca), P, magnesium (Mg) and potassium (K) contents (2771.07, 5024.56, 3501.36 and 5101.99 mg/kg, respectively) in the grain and significantly higher trace mineral content for Copper (Cu), manganese (Mn), zinc (Zn) and iron (Fe) (5.95, 23.71, 49.97 and 147.01 mg/kg, respectively). In addition, secondary metabolites, specifically anthocyanins, were shown to be abundant in grains (35.92 mg/kg), which reinforces its potential in increasing the immunity. Amaranth grains also have high content of starch (65–75%), proteins (15.4–16%), lipids (6.98–7.22%) and several vitamins [41], besides being gluten free, making it important for celiac patients [42]. Bearing in mind the importance of recycling the nutrients contained in the sewage, the present study examined the effectiveness of sewage treatment by-products as organic fertilizer in the culture of *Amaranthus cruentus* L. Specifically, the study analyzed the responses of plants to sanitary sewage treatment by-products in terms of growth, shoot biomass production, photosynthetic pigments, leaf gas exchange and leaf nutrients. It was hypothesized that the addition of sanitary sewage treatment by-products may positively influence the performance of amaranth plants thus resulting in reduced fertilizer costs, depletion of natural resources and avoiding environmental pollution. The study could provide insights into the potential benefits and limitations of this approach.

## Materials and methods

### Obtaining organic fertilizers

**Collection of previously treated sewage and and obtaining biosolid.**   The Faculty of Engineering—Bauru at Universidade Estadual Paulista (UNESP) currently maintains a collaborative agreement (code 2100.0632) with the Department of Water and Sewage of Bauru (DAE), which is a foundation of the Municipal Government of the Bauru city. This agreement was officially published in the gazette on May 2, 2023, and is in effect until May 18, 2028. The primary objectives of this agreement refer to the development, dissemination, and practical application of research with the ultimate goal of benefiting society. Under this agreement, researchers are granted access to the sanitary sewage collection site for their research purposes.

Furthermore, they are authorized to collect and employ sanitary sewage for their research purposes. Notably, for the cultivation of microalgae and sunflowers, there is no need for legal or institutional permissions, as this activity is an integral component of the researchers' ongoing research activities.

The sanitary sewage previously anaerobically treated used for cultivation of microalgae was collected at the STP in the district of Tibiriçá, in Bauru/Sao Paulo (22˚13'38.7"S 49˚1 2'47.0"W). The anaerobic treatment took place in a anaerobic upflow filter, with a volume of 87.30 $m^3$ and a hydraulic detention time (HDT) of 5.05 h. The sludge was obtained in the third chamber of the baffled anaerobic reactor, which had three chambers with a total volume of 597 L and HDT of 8 h. After collecting, the sludge was oven dried at 60˚C. Then, it was crushed and ground in a mill to facilitate the absorption by the plants.

**Microalgae cultivation.** The microalgae were cultivated in a transparent box with 50 L of anaerobically pre-treated sewage with 5 L of inoculum of native microalgae from sanitary wastewater. The cultivation of microalgae took place at room temperature under a shading fabric that blocked 50% of natural light radiation (Solpack), which according to Ruas et al. [43] is a method of preventing photoinhibition and favoring the diversity of the reactor community without compromising the disinfection capacity. After this period, the effluent was transferred to 200 L drums. The microalgae cultivation took place between the months of January and April (summer/fall), with an average temperature of 28˚C, and the average light intensity of 919.48 $\mu$mol m$^{-2}$ s$^{-1}$. In the current investigation, the most prevalent genera of microalgae identified were colonial cyanobacteria, specifically *Aphanotece* sp. and *Aphanocapsa* sp., along with filamentous algal species. Following these, *Chlorella* sp., *Scenedesmus* sp., and *Chlamydomonas* sp. were also present. The microalgae utilized in this study were sourced through a bioprospecting process, which enabled the selection of species naturally occurring in sewage. Consequently, these microalgae were already well-adapted to the culture medium, facilitating their growth and development throughout the experimental process [44]. Eleven weeks of cultivation were necessary to obtain the amount of microalgae established for supplementation of the plants.

**Sedimentation of microalgae, obtaining reclaimed water and biomass from microalgae.** After filling the decantation drums with the microalgae together with the effluent, 100 mg.L$^{-1}$ of tannin (Tanfloc) were added for the coagulation, flocculation and sedimentation of the microalgae [45]. After 24 hours the addition of tannin, all the reclaimed water and the sedimented microalgae were collected.

## Plant material and growth conditions

Seeds of *Amaranthus cruentus* L. cv. BRS Alegria were used. This cultivar was developed by the 13th Cerrados Agricultural Research Center, originated from the strain *A. cruentus* AM 5189 from United States [46]. The experiment was carried out in the greenhouse at the Department of Biological Sciences at the Sao Paulo State University/Bauru Campus, under natural photoperiodic conditions and minimum and maximum average temperature of 17 and 33˚C, respectively. The seeds were sown in 4 L plastic pots filled with vermiculite. After one week, the plants were thinned to one per pot. Each pot received 250 mL of 20% or 70% full strength Long Ashton (LA) solution [47] from sowing and continued three times a week.

## Dried and fresh microalgae application

After sedimentation, part of the microalgae was kept in a refrigerator before using and the other was distributed in plastic trays for drying in a forced air circulation oven at 65˚C for 48 hours. The fresh and the dry microalgae were applied in two periods: 20 days before sowing

and 24 days after sowing. The plants were divided into two main groups: (I) with or without dry microalgae or (II) with or without fresh microalgae. Each group was divided into five subgroups:

1 Group I: (1) 70% LA; (2) 20% LA; (3) 20% LA + 2 g of dry microalgae; (4) 20% LA + 4 g of dry microalgae and (5) 20% LA + 8 g of dry microalgae.

2 Group II: (1) 70% LA; (2) 20% LA; (3) 20% LA + 3 g of fresh microalgae; (4) 20% LA + 6 g of fresh microalgae and (5) 20% LA + 9 g of fresh microalgae.

### Reclaimed water application

The reclaimed water obtained after separating the microalgae was kept in a refrigerator before using as a plant nutritive solution. The plants were divided into three groups: (1) 70% LA; (2) 20% LA and (3) reclaimed water. The reclaimed water was applied in the same volume as the Long Ashton solution, three times a week.

### Biosolid application

Two biossolid applications were made: the first 20 days before sowing and the second about 20 days after sowing. The plants were divided into six groups: (1) 70% LA; (2) 20% LA; (3) 20% LA + 3 g of biosolid; (4) 20% LA + 6 g of biosolid; (5) 20% LA + 10 g of biosolid and (6) tap water + 10 g of biosolid.

### Growth measurements

At approximately 7 day intervals, the height (cm), the diameter (mm) of the stem at insertion of cotyledons and the number of visible leaves were measured. A millimeter ruler was used for height measurements and a digital caliper (KingTools) was used for stem diameter measurements.

### Gas exchange measurements and photosynthetic pigments determination

An infrared gas analyzer (LCpro Portable Photosynthesis System, ADC Bioscientific Ltd., Hoddesdon, UK) was used for measurements of photosynthesis ($A$), stomatal conductance ($g_s$), transpiration ($E$) and $CO_2$ concentration in the substomatic cavity ($C_i$), in the youngest fully expanded leaf after 44 days of cultivation. The measurements were carried out inside the greenhouse between 8 am and 10 am under ambient conditions of temperature, partial pressure of $CO_2$ and water vapour pressure of the air. Photosynthetically active radiation (PAR) of 1000 $\mu$mol m$^{-2}$ s$^{-1}$ was supplied by a lamp coupled on the top leaf chamber. The leaf was kept under this PAR until a steady-state rate was achieved. Total chlorophyll and carotenoids were extracted from leaf discs of known area in 80% aqueous acetone according to [48] and the concentration expressed on a leaf area basis (g m$^{-2}$).

### Dry mass determination

At the end of the experimental period, plants of each treatment were selected randomly for dry matter determinations. The plants were divided into stem and leaves before been oven dried at 60°C for 48 h and the dry matter was expressed as g plant$^{-1}$.

### Chemical analysis of microalge, reclaimed water, biosolid and amaranth leaves

The analyses were done in the Laboratory of Soil, located in the Department of Soils and Environmental Resources at the Faculty of Agricultural Sciences/UNESP/Botucatu/SP. Chemical leaf nutrients status was determined in oven dried and finely ground leaf by the semi-micro-Kjeldahl method, after sulphurous digestion according to [49]. The determination of macronutrients (N, P, K, Ca, Mg and Sulfur (S)) and micronutrients (Cu, Fe, Mn, Zn and sodium (Na)) of dry and wet microalgae, biosolid and reclaimed water was performed according to the methodology of the Manual of Official Analytical Methods for Fertilizers and Correctives.

### Concentration of heavy metals and physicochemical characteristics of anaerobically digested sewage

The analysis of physicochemical parameters and heavy metals of anaerobically digested sewage by UAF (Upflow anaerobic Filter) and BAR (baffled anaerobic reactor) was conducted in the laboratory following a methodology described in Standard Methods for the Examination of Water and Wastewater [50]. In S1 Table describes the physiochemical composition of the sanitary sewage post-treated by the anaerobic upflow filter (UAF) and the baffled anaerobic reactor (BAR) while the concentrations of heavy metals in the sewage treated by UAF and ABR are presented in S2 Table. There was no significant difference for most parameters of the two raw anaerobic sewage.

### Statistical analysis

The data were submitted to simple analysis of variance (ANOVA) by using the software SPSS/PC 9.0 for Windows. Statistical analysis was applied to each group of plants separately. Quantitative changes in the different variables were analyzed using a test of multiple comparison to determine differences between treatments at 5% significance level. The least significant difference (LSD) test was used for assumed equal variances and the Games-Howell test for non-assumed equal variances. The statistical analyzes for the number of leaves were performed after transforming the data into $\sqrt{x + 0.5}$.

## Results

### Growth and dry matter production

In this study, four different biofertilizers from the treatment of sewage were applied in plants grown under 20% full strength Long Ashton (LA) solution in order to verify the potential of these biofertilezer in supplying nutrients for growth of amaranth plants (Fig 1). In well nourish plants (70% LA), the plant height, stem diameter and number of leaves were significantly higher than under lower strength nutrient solution (20% LA) at the end of experimental period. From 42 days after sowing, the number of leaves produced in the plants grown under 70% LA practically remained constant until the end of the experiment due to the beginning of flowering. Comparing the different applications of dry microlagae with 20% LA after 28 days of sowing, it can be seen that the plants supplied with 4 g of microalgae presented significantly lower height while the other did not differ (Fig 1A). From 35 days after sowing and until the end of the experiment, the plants supplied with 4 g and 8 g dry microalgae presented similar values in height, reaching at the end of experiment 48.5 and 51.5 cm, respectively, significantly surpassing 20% LA, which reached 40.3 cm, and the 2 g dry microalgae which reached 39.6 cm (Fig 1A). The number of leaves produced per plant began to differ significantly between treatments after 28 days of sowing (Fig 1B). The plants grown under 8 g of dry microalgae reached

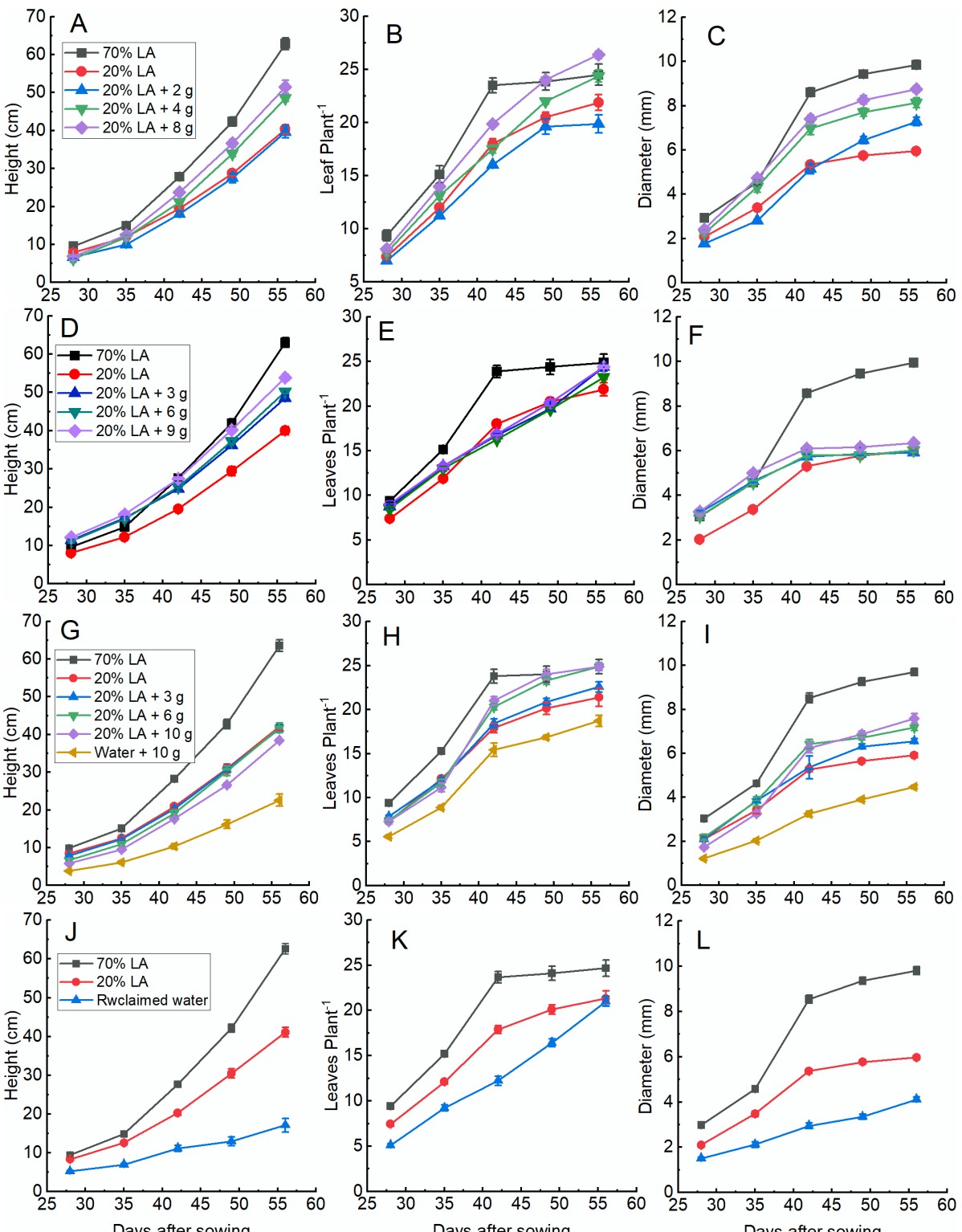

**Fig 1.** Plant height (A, D, G and J), number of leaf per plant (B, E, H and K) and stem diameter (C, F, I and L) of amaranth plants grown under addition of dry microalgae (A, B and C), wet microalgae (D, E, and F), biosolid (G, H and I) or reclaimed water (J, K and L) during the growth period. Values are mean of 8 plants.

the number of leaves produced in 70% LA 49 days after sowing, with both treatments presenting an average of 24 leaves per plant (Fig 1B). After 56 days of sowing, the number of leaves in 8 g of dry microalgae exceeded the number of leaves of 70% LA, reaching 26 leaves per plant. The stem diameter of the plants grown under supplementation with dry microalgae at the beginning of the experiment were very similar to the stem diameter of the 20% LA. At the end of experiment, the stem diameter of the of 4 g and 8 g of dry microalgae showed similar values, 8.1 and 8.7 mm, respectively, values significantly higher than the stem diameter of 20% LA and 2 g of dry microalgae (Fig 1C).

In Fig 1D, it is noted that in the first two weeks of measurements the plants supplied with wet microalgae (3 g, 6 g and 9 g) presented heights greater than the two controls (20% and 70% LA). From 42 days after sowing, the treatments with wet microalgae resulted in heights greater than the treatment with 20% LA, but lower than the treatment with 70% LA, with the treatment with 9 g of wet microalgae showing the highest height among the treatments with wet microalgae. Regarding the number of leaves produced, the treatments with wet microalgae, between 28 and 35 days after sowing and at the end of the experiment, showed a slightly higher number than 20% LA (Fig 1E). At the end of the experiment, the number of leaves in plants grown with the addition of 3 g and 9 g of wet microalgae practically equalled that of 70% LA. The diameter of stem in plants grown under addition of wet microalgae had values higher than the 20% LA control and similar values to the control 70% LA until about 35 days after planting (Fig 1F). After this period, the treatments with wet microalgae were not significantly different from 20% LA.

Plants grown under 3 g and 6 g of biosolid plus 20% LA did not differ significantly from the 20% LA control in terms of height of the plants over the period of growth (Fig 1G). However, the supply of 10 g of biosolid with 20% LA or with 10 g of pure biosolid showed a reduction of 10% and 47% at the end of growth period in relation to the 20% control, respectively. Between 28 and 35 days after sowing, the number of leaves per plant did not differ significantly between the control 20% LA and the doses of 3 g, 6 g and 10 g of biosolid plus 20% LA (Fig 1H). However, plants grown with only 10 g of biosolid showed lower values than the other treatments. At the end of the experiment, plants with 6 g and 10 g of biosolid plus 20% LA had significantly higher number of leaves than the 20% LA control and 3 g of biosolid plus 20% LA and the pure biosolid, but equal to the 70% control. After 28 days of sowing, the application of biosolid at doses of 3 g and 6 g per pot did not result in a significant change in stem diameter compared to 20% LA (Fig 1I). However, application of 10 g of biosolid with 20% LA or 10 g of biosolid with tap water resulted in a significant reduction in stem diameter of 18% and 42%, respectively. At the end of the experiment, there was a positive correlation between the biosolid doses applied together with 20% LA and the stem diameter. However, the application of biosolid with tap water presented a reduction of approximately 24% in the stem diameter compared to 20% LA.

At 28 days after sowing, the height of plants treated only with reclaimed water reached approximately 62% and 55% of the two controls 20% LA and 70% LA, respectively. At the end of the experimental period, the height of plants under reclaimed water presented much lower percentages (about 42% and 27%, respectively). At the beginning of the experiment, the average number of leaves in the reuse water was 5.1, representing about 69% and 54% of the controls 20% LA and 70% LA, respectively (Fig 1K). Despite the reclaimed water having initially affected the production of leaves, at the end of the experiment the production was equal to the control 20% LA. This was due to the fact that the plants under 20% stopped producing new leaves because they entered the reproductive phase earlier than the plants in the reclaimed water. The diameter of stem 28 days after sowing represented 72% and 50% of the two controls, respectively (Fig 1L). At the end of experimental period, the diameter of stem of the

plants treated only with reclaimed water reached 69% and 42% of the two controls 20% LA and 70% LA, respectively.

Dry mass production in the shoot under 4 g and 8 g of dry microlagae significantly exceeded the 2 g and 20% LA (Fig 2A). The plants under 8 g of dry microalgae produced more than twice the dry mass produced in 20% LA, and produced only 30% less compared to 70% LA. The dry microalgal treatments resulted in a higher dry mass investment into leaves when compare with the two controls. On the contrary, 70% and 20% LA showed greater investment of dry mass in the stem, with investment of 44.2 and 46.4% of dry mass in this organ, respectively, while the treatments with microalgae did not exceeded 43%. The investment in inflorescence in 70% LA was the highest with 9.2%, in the other treatments was between 5.6% and 6.6%. In Fig 2B, it can be seen that the increase in the amount of wet microalgae applied resulted in an increase in the amount of dry mass of shoot, being greater than the mass obtained by 20% LA. However, the observed increase from the 3 g to 6 g of microalgae was not significantly different. The application of 9 g of wet microalgae stands out among them, presenting 40% more dry mass when compared to 20% LA, and an increase in the dry mass of stem, leaves and inflorescence can be observed.

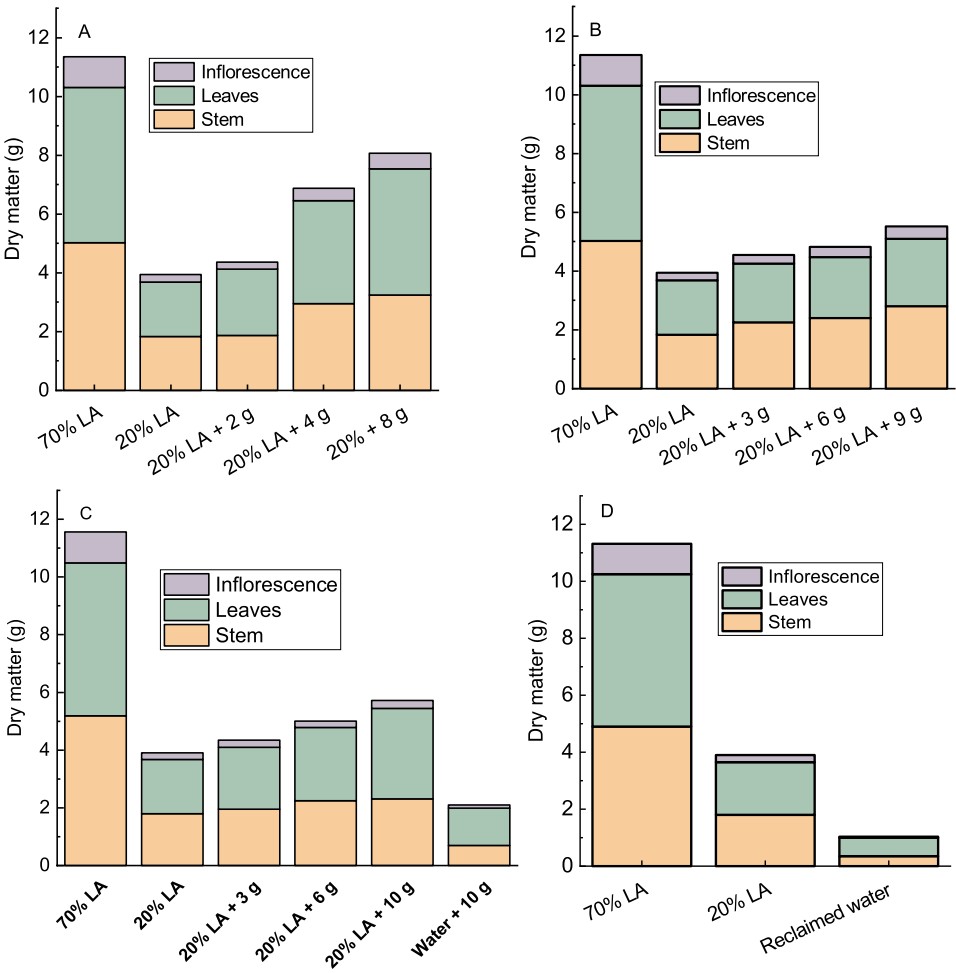

**Fig 2. Dry matter partitioning in the above ground of amaranth plants grown under addition of dry microalgae (A), wet microalgae (B), biosolid (C) or reclaimed water (D) after 56 days of sowing.** Values are mean of 8 plants.

Application of 3 g of biosolid did not result in a significant increase in shoot dry mass compared to 20% LA (Fig 2C). However, application of 6 and 10 g plus 20% LA resulted in a significant increase of 28% and 47%, respectively, when compared to 20% LA. It is important to note that the biosolid alone was able to produce about 54% of the dry mass of shoot of the 20% LA control, and about 18% of the 70% LA control. In relation to dry matter partitioning in the above ground, the addition of increasing doses of biosolid resulted in less investment of dry mass into stem from 45% for 3 g of biosolid plus 20% LA to 33% under pure biosolid when compared to 20% control (Fig 2C). Contrary results were found for the investment in dry mater in the leaves. The investment in dry matter in leaves was from 49% to 61% when compare with 20% LA. However, the addition of biosolid did not result in significant alteration in the proportion of the dry matter investment in inflorescence.

At the end o experiment, the plants grown under only reclaimed water produced 26.5% of the dry mass of the aerial part produced by the control 20% LA, and of the total mass produced about 64% is represented by the leaves, 33.5% by the stem and 2.6% by the inflorescence (Fig 2D). In the 20% LA control, the investment in dry mass in stem and leaves was similar (48% and 46%, respectively). In relation to the 70% LA control, the plants supplied only with reclaimed water produced about 9% of the dry matter of the above ground.

## Photosynthetic pigments and Gas exchange characteristics

Plants grown under 20% LA showed a reduction of about 30% in total chlorophyll and total carotenoids compared to 70% LA (Table 1). The content of total chlorophyll was significantly lower in plants supplied with 4 g of dry microalae when compared with 20% LA and with the application of 8 g of dry microalgae. The total carotenoids was significantly reduced under 20% LA without or with the addition of microalgae when compared to 70% LA. However, the addition of 8 g of microalgae resulted in an increase in total carotenoids compared to 20% LA

**Table 1. Total chlorophyll and carotenoids (g m$^{-2}$) of amaranth leaves.** Dm, dry microalgae; Wm, wet microalgae. Mean values sharing the same letter within each treatment group do not differ significantly at P = 0.05.

| Treatments | Chlorophyll | Carotenoids |
|---|---|---|
| 70% LA + 0 g | 0.270 ± 0.016 a | 0.081 ± 0.003 a |
| 20% LA + 0 g | 0.190 ± 0.007 b | 0.058 ± 0.002 c |
| 20% LA + 2 g Dm | 0.179 ± 0.016 bc | 0.057 ± 0.003 c |
| 20% LA + 4 g Dm | 0.157 ± 0.013 c | 0.053 ± 0.000 c |
| 20% LA + 8 g Dm | 0.218 ± 0.008 ab | 0.069 ± 0.002 b |
| 70% LA + 0 g | 0.270 ± 0.016 a | 0.081 ± 0.003 a |
| 20% LA + 0 g | 0.190 ± 0.007 b | 0.058 ± 0.002 b |
| 20% LA + 3 g Wm | 0.143 ± 0.005 c | 0.045 ± 0.001 c |
| 20% LA + 6 g Wm | 0.145 ± 0.005 c | 0.046 ± 0.001 c |
| 20% LA + 9 g Wm | 0,141 ± 0.008 c | 0.046 ± 0.002 c |
| 70% LA + 0 g | 0.270 ± 0.016 a | 0.081 ± 0.003 a |
| 20% LA + 0 g | 0.190 ± 0.007 c | 0.058 ± 0.002 bc |
| 20% LA + 3 g biosolid | 0.153 ± 0.012 c | 0.045 ± 0.003 d |
| 20% LA + 6 g biosolid | 0.188 ± 0.024 cd | 0.054 ± 0.006 cd |
| 20% LA + 10 g biosolid | 0.236 ± 0.012 ab | 0.067 ± 0.003 b |
| Water + 10 g biosolid | 0.218 ± 0.006 bd | 0.066 ± 0.001 b |
| 70% LA + 0 g | 0.270 ± 0.016 a | 0.081 ± 0.003 a |
| 20% LA + 0 g | 0.190 ± 0.007 b | 0.058 ± 0.002 b |
| Reclaimed water | 0.240 ± 0.014 a | 0.067 ± 0.004 b |

without microalgae, with no alteration in the other treatments. On the other hand, addition of wet microalgae was not able to maintain the same content of photosynthetic pigments as the 20% LA without wet microalgae.

Addition of 3 g and 6 g of biosolid did not result in a significant change in total chlorophyll content when compared with the 20% LA control (Table 1). Otherwise, application of 10 g plus 20% LA and 10 g of pure biosolid resulted in a significant increase of 22% and 13%, respectively, in total chlorophyll content when compared to the 20% LA control. The total chlorophyll content in the treatment with 10 g of biosolid plus 20% LA reached a value similar to the control 70% LA. Application of 3 g of biosolid resulted in lower carotenoids content compared to 20% LA control, 10 g biosolid plus 20% LA and 10 g of biosolid plus tap water. However, the other biosolid treatments did not differ from the 20% LA control.

Plants supplied only with reclaimed water showed a significant increase of 24% in the total chlorophyll content in relation to the 20% LA (Table 1). Comparing the plants under reclaimed water with the plants under 70% LA, application of reclaimed water resulted in about 87% of the total chlorophyll content found in the control 70% LA. The total carotenoids of plants grown under reclaimed water were similar to 20% LA.

The rate of photosynthesis ($A$) was significantly reduced under 20% LA compared to 70% LA (Fig 3A). Addition of 2 g of dry microalgae induced an increase in $A$ above the values found in plants under 20% LA, becoming equal to the 70% LA. Comparing $A$ of the plants grown under addition of dry microalgae with the 20% LA, only the addition of 2 g and 8 g of dry microalgae showed an increase in $A$. Plants grown under 20% LA resulted in a significant increase in stomatal conductance ($g_s$) compared with 70% LA (Fig 3B). However, addition of 2 g and 4 g showed lower $g_s$ than the control 20% LA while under 8 g of dry microalgae the $g_s$ was higher than in 20% LA. The concentration of $CO_2$ in the substomatal cavity ($C_i$) was significantly lower in plants grown under 70% LA compare to 20% LA (Fig 3C). The addition of 4 g and 8 g of dry microalgae did not alter the $C_i$ compared to 20% LA. However, addition of 2 g of dry microalgae resulted in lower $C_i$. The addition of 2 g of microalgae resulted in values of $C_i$ similar to 70% LA. The treatments with wet microalgae showed a lower $A$ than the controls 20% LA, with 3 g having the lowest rate (Fig 3D). The rate of $g_s$ also presented values below the two controls for the treatments with wet microalgae (Fig 3E). Despite the $g_s$ being smaller in plants grown under addition of wet microalgae than the in the 20% LA, the $C_i$ it was higher or equal to the control 20% LA (Fig 3F).

Application of 3 g and 6 g of biosolid resulted in no change in $A$ when compared to 20% LA control (Fig 3G). However, application of 10 g with or without 20% LA resulted in an increase in $A$, surpassing the $A$ values of the 70% LA control. Therefore, the addition of 10 g of biosolid promoted $A$ when compared to the two controls. Biosolid application with or without 20% LA resulted in a significant decrease in $g_s$ when compared to 20% LA control (Fig 3H). Among the biosolid doses there was a positive correlation between biosolid dose and $g_s$. Application of 10 g of biosolid with tap water resulted in $g_s$ values similar to the 70% LA control, while the $g_s$ in the 20% LA control was higher than in all other treatments. The application biosolid at the different doses resulted in a reduction in $C_i$ when compared to 20% LA and 70% LA controls (Fig 3I). It is important to note that although the $g_s$ increased with the increase in the biosolid doses, the $C_i$ was reduced.

Although the reclaimed water does not have an ideal balance of nutrients, the value of $A$ was 17.21 $\mu$mol m$^{-2}$ s$^{-1}$, a reduction of 20% and of 33% when compared to 20% and 70% LA, respectively (Fig 3J). Plants grown under reclaimed water presented a significant reduction in $g_s$ of 37% compared to 20% LA (Fig 3K). However, when compared to 70% LA the reduction was only 20%. The $C_i$ for the plants grown under reclaimed water did not differ significantly from the 20% control (Fig 3L).

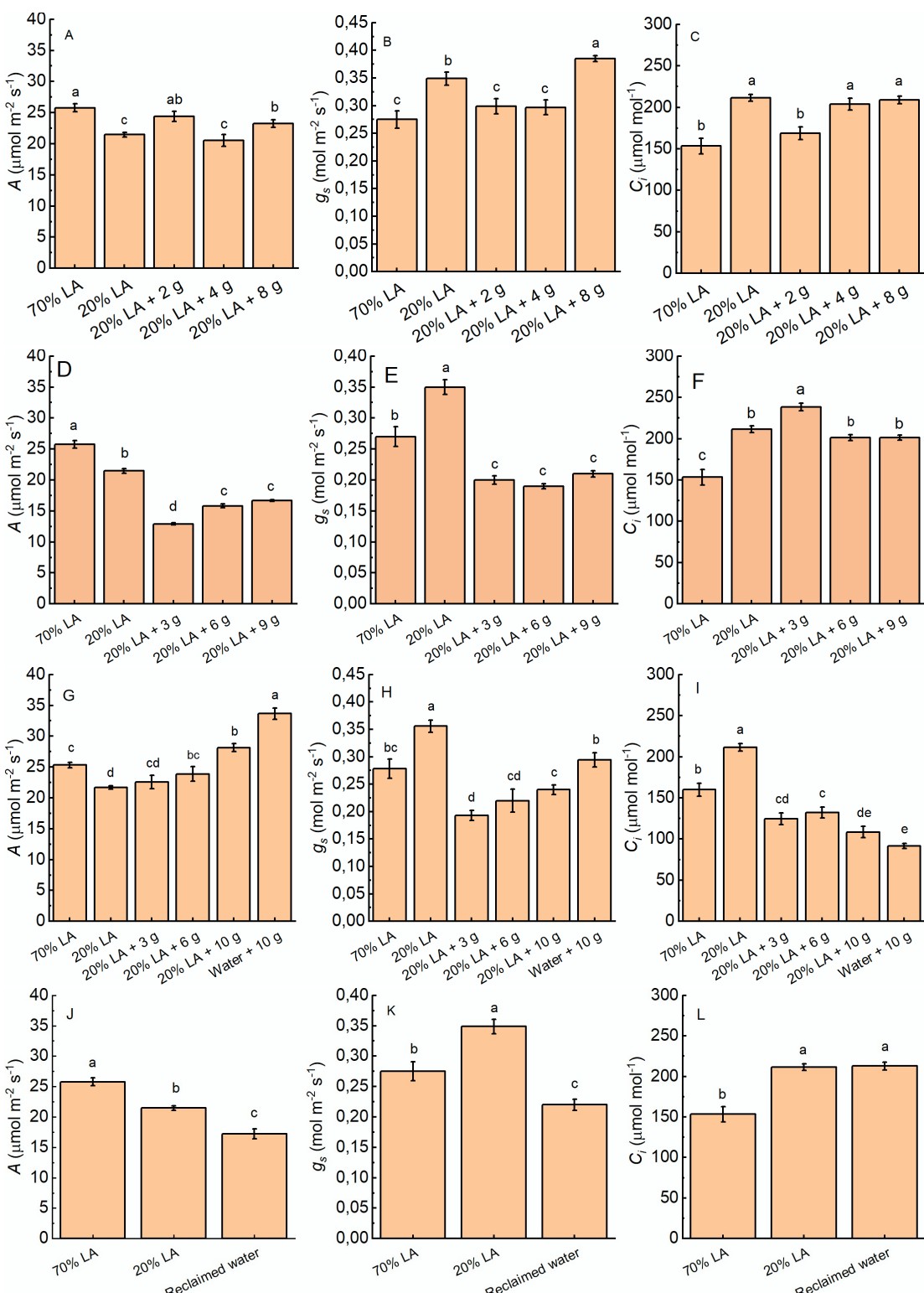

**Fig 3.** *A*; A, D, G, J), stomatal conductance ($g_s$; B, E, H, K), and intercellular $CO_2$ concentration ($C_i$, C,F, I, L) of amaranth plants grown under addition of dry microalgae (A, B, C), wet microalgae (D, E, F), biosolid (G, H, I) or reclaimed water (J, K, L) after 43 days of sowing. Values are mean±SE of 8 plants.

## Chemical composition of biofertilizers and leaf of amaranth

The characterization of the dry microalgae used as a fertilizer in this experiment shows that its nutrient content is unbalanced in relation to plant needs (Table 2). The biomass of these microalgae has an adequate content of N, Ca, P and S, while K, Mg and Mn had a concentration below the requirements of a proper amaranth growth. In terms of micronutrients, the imbalance is shown by the content of Na, Fe, Zn and Cu that are above the appropriate level. The analysis of wet microalgae shows lower levels of macronutrients compared the content of macronutrients found in the leaves of well nourish plants (Table 2). In relation to micronutrients content, Mn presented a lower level while Cu, Fe and Zn were found in higher concentration. The content of sodium (Na) was within ideal levels for amaranth.

The chemical analysis of the biosolid reveal that the macronutrients content of N, K, Mg and S are below the suitable levels when compared with the plants grown under optimal availability of nutrients (70% LA), while the content of P and Ca are above the suitable level, respectively (Table 2). The content of micronutrients Cu, Fe, Zn and Na are above the level found in the plants grown under suitable nutrient supply (70% LA). On the contrary, the content of Mn was lower. In the Table 2 the chemical analysis of reclaimed water shows that it contains significant amounts of N, K, Ca and S after the wastewater treatment with microalgae but lower amounts of P and Mg. In relation to micronutrients, these are found in significantly lower concentrations.

As control treatments, we used plants grown under supply of 70% or 20% full strength LA nutrient solution in the absence of microalgae (70% LA and 20% LA). In previous experiments it was observed that desirable plant growth was achieved with the application of the 70%

**Table 2. Nutrient content of biofertilizers from wastewater treatment and leaves of amaranth plants.** Dm, dry microalgae; Wm, wet microalgae; BDL, below detection level.

| Variables | Macronutrients (g Kg$^{-1}$) | | | | | | Micronutrients (mg Kg$^{-1}$) | | | | | |
|---|---|---|---|---|---|---|---|---|---|---|---|---|
| | N | P | K | Ca | Mg | S | B | Cu | Fe | Mn | Zn | Na |
| Biofertilizers | | | | | | | | | | | | |
| Dry microalgae (Dm) | 57 | 14.2 | 5.2 | 6.8 | 3.1 | 4.0 | - | 202 | 5572 | 102 | 2925 | 20803 |
| Wet microalgae (Wm) | 5.6 | 1.3 | 0.1 | 0.4 | 0.2 | 0.4 | - | 27.7 | 510 | 13 | 291 | 164 |
| Biosolid | 24.4 | 13.3 | 0.9 | 8.4 | 2.2 | 2.8 | - | 177 | 24752 | 115 | 3167 | 858 |
| Reclaimed water (mg L$^{-1}$) | 49.0 | 2.0 | 151 | 13 | 2.0 | 9.4 | 0.19 | BDL | 0.08 | BDL | 0.01 | 47.7 |
| Amaranth leaves | | | | | | | | | | | | |
| 70% LA + 0 g | 28 | 6.4 | 60 | 7 | 13.9 | 3.9 | 24 | 15 | 229 | 359 | 36 | 170 |
| 20% LA + 0 g | 20 | 4.6 | 39 | 7 | 18.7 | 4.9 | 35 | 9 | 115 | 252 | 25 | 160 |
| 20% LA + 2 g Dm | 23 | 3.8 | 48 | 5 | 20.0 | 6.0 | 27 | 10 | 93 | 401 | 37 | 120 |
| 20% LA + 4 g Dm | 25 | 5.2 | 39 | 7 | 19.8 | 3.9 | 36 | 9 | 82 | 98 | 39 | 140 |
| 20% LA + 8 g Dm | 29 | 6.4 | 42 | 6 | 21.1 | 4.0 | 39 | 11 | 89 | 103 | 43 | 190 |
| 20% LA + 3 g Wm | 22 | 5.1 | 35 | 11 | 24.1 | 5.4 | 42 | 10 | 86 | 152 | 27 | 180 |
| 20% LA + 6 g Wm | 22 | 5.0 | 36 | 12 | 24.9 | 4.8 | 45 | 9.0 | 78 | 176 | 30 | 180 |
| 20% LA + 9 g Wm | 20 | 4.6 | 32 | 11 | 23 | 5.0 | 45 | 11 | 75 | 186 | 43 | 170 |
| 20% LA + 3 g biosolid | 21 | 4.9 | 37 | 9 | 20.2 | 4.3 | 42 | 13 | 84 | 239 | 105 | 140 |
| 20% LA + 6 g biosolid | 22 | 4.7 | 45 | 8 | 18.1 | 3.7 | 38 | 14 | 101 | 252 | 126 | 130 |
| 20% LA + 10 g biosolid | 25 | 4.9 | 46 | 7 | 16.0 | 3.0 | 41 | 17 | 138 | 221 | 198 | 140 |
| Water + 10 g biosolid | 20 | 3.4 | 32 | 7 | 16.2 | 6.8 | 28 | 17 | 142 | 224 | 222 | 120 |
| Reclaimed water | 26 | 6.3 | 44 | 9 | 26.2 | 3.7 | 25 | 3 | 90 | 37 | 2 | - |

- not determined

nutrient solution [51]. Leaves are plant organs that best reflect the nutritional status of the plant. Therefore, they indicate changes in the availability of nutrients in the soil. The chemical analyses of nutrients in the leaves were done 56 days after sowing. Plants grown under 20% LA showed lower levels of almost all nutrients analysed compare to 70% LA (Table 2). The leaf chemical analysis show that of all the doses of dry microalgae used, the best results were found in the dose of 8 g, which showed superior results than the 20% LA for most macronutrients, except for S. The contents of micronutrients boron (B), Cu, Zn and Na were increased under the application of 8 g of dry microalgae compared to 20% LA. It is interesting to note that even the content of Cu, Fe, Zn and Na were higher in the dry microalgae, no excess of this nutrients were found in the leaves of amaranth. There was no clear correlation between the dose of wet microalgae applied and the concentration of nutrients observed in the leaves. Application of 9 g of wet microalgae resulted in higher levels of Ca, Mg, B, Cu, Zn and Na in the leaves of amaranth when compared to 20% LA (Table 2). When compared to the 70% control, the values for Ca, Mg, B and Zn were higher in plants with 9 g of wet microalgae. On the other hand, the content of K, Fe and Mn were lower than the two controls.

The content of N, K, B, Cu and Zn in the leaves of amaranth under the addition of 6 and 10 g of biosolid plus 20% LA were increase compare to 20% LA while P and Ca did not change significantly (Table 2). The content of Mg and S were decreased with the increase in the doses of biosolid compared with 20% LA. Although the content of Na was high in the sludge its content was reduced to below the content of 20% LA. The Mn content under 10 g of biosolid with or without LA were reduced. When 10 g of biosolid were applied with tap water, the content o P, K, Mg, B and Mn were decreased while the content of S was increased. The high content of Cu, Fe and Zn in the biosolid resulted in more concentration in the leaves compared with the control 20% LA. Although the supply of Na in the biosolid was high, this did not result in a further increase in the concentration in the amaranth leaves. The leaves under reclaimed water presented contents of N, P, K, Ca and Mg that were higher than in the 20% LA (Table 2). However, the levels of micronutrients were much lower. Comparing reuse water with 70% LA, the leaves from plants grown under reclaimed water had lower concentrations of N and K but higher concentrations of Ca and Mg.

## Discussion

Various methods to replace chemical fertilization have been explored, including the use of secondary products from different stages of sanitary sewage treatment that provide a rich source of nutrients. This study focuses on the use of sludge generated from anaerobic treatment of sanitary sewage, microalgae biomass as post-treatment of sanitary sewage, and reclaimed water obtained after separating microalgae as alternatives for recovering and using nutrients as fertilizers in amaranth cultivation.

The generation and appropriate disposal of sludge produced during the treatment of sanitary sewage has been a topic of concern, with efforts to reduce its production [11] and to find sustainable reuse options. In Brazil, for instance, the annual production of dry matter from sludge exceeds 150,000 tons [52]. Fertilization has been identified as a promising destinations found for the sludge, with laws and regulations in place to ensure safety parameters for its application as biosolids [13, 53]. The sludge and microalgae are rich sources of nutrients, particularly nitrogen, phosphorus, and micronutrients [27, 54, 55]. Moreover, the application of microalgae in soil can improve soil organic matter, which can enhance plant productivity [6, 56, 57]. Combining microalgae cultivation with sanitary sewage treatment can improve the cost-benefit of both processes [58]. The treated effluent obtained after the cultivation of microalgae is a crucial aspect of sanitary sewage treatment, and its use in agricultural practices can

reduce the use of potable water for irrigation and help to recycle remaining nutrients that would otherwise be released into water bodies.

The nutrient composition of the organic fertilizers used in this study revealed that they all contained nutrients essential for plant development, albeit at different concentrations depending on the stage of treatment. The anaerobic sludge has lower levels of macronutrients such as N and K, compared to the dry microalgae biomass. However, the N and Ca content of the biosolid were suitable for the requirements of amaranth. The metal analysis showed that due to its high Zn concentration, the biosolid used was classified as class 2 by CONAMA resolution Nº 498/2020, which restricts the maximum annual application rate based on the soil's characteristics [13]. Similar to studies by [59, 60], the accumulation of Zn in the leaves of plants that received biosolids occurred together with copper accumulation. Zinc toxicity can reduce the leaf nutrient content, particularly for K, Fe and Mg, by inhibiting assimilation and interfering with nutrient translocation to the leaves [61–63]. The high Zn concentration in the biosolid might have affected the assimilation and/or translocation of Fe, which was present in high concentrations in the biosolid but did not accumulate in the leaf tissues, it might have also influenced the leaf Mn content that did not exceed 20% the content in the treatment.

The nutrient composition of microalgae biomass depends on various cultivation conditions, such as temperature, luminosity, and sewage characteristics. These factors influence the composition of microalgae community, and different species of microalgae have different capacities for nutrient removal [64–66]. Additionally, the native species of sanitary sewage are well-adapted to that environment, and changes in the community can occur based on different seasons [67, 68]. While microalgae have shown to have nutrients available for the development of amaranth, wet microalgae contains smaller amounts of these nutrients. Despite the high concentration of micronutrients such as Cu, Fe, Zn and Na in the dry microalgae biomass and in the sludge, there was no significant increase in the leaves. This may indicate that the plant has specific mechanisms to limit the absorption of excess nutrients and avoid physiological damage [69] or the presence of interactions between different nutrients [70, 71]. For example, in the plants that received the biosolid, the high concentration of Zn in the microalgae biomass may have interfered with the assimilation and translocation of nutrients such as Mn and Fe, leading to leaf concentrations lower than the 20% control. It is possible that the foliar concentration of Zn was not high in these treatments, but the inhibition could have occurred in the roots, as demonstrated by [72] where Zn and Cu showed a marked accumulation in the roots in relation to the shoot, which represents a plant strategy to mitigate toxicity by these metals.

Saline stress conditions can induce nutrient deficiencies in leaves, such Fe and Mn [73]. Additionally, this condition is often accompanied by an increase in foliar Na concentration and a decrease in K and N [74]. However, in the present study, with 8 g of dry microalgae and the treatments with wet microalgae showed an elevated concentration of Na in relation to the control treatments.

There were no other signs of saline stress conditions, indicating that the nutrient deficiencies found in this study may not be related to salt stress conditions. It is noteworthy that Na is considered an essential nutrient in $C_4$ plants and is involved in various physiological processes, such as the recovery of phosphoenolpyruvate and the synthesis of chlorophyll [75–77]. For a more precise investigation in relation to the stress caused by nutrition, proline analysis could be a good indicator, as it has been observed that an increase in proline production is related to stress situations [78, 79]. Another possibility is that the micronutrients may not have been fully released. According to [80], microalgae can be used as a slow-release organic fertilizer, releasing nutrients gradually. However, the mechanism of nutrient release by microalgae requires further investigation.

Reclaimed water, in turn, due to the efficiency of nutrient removal by microalgae, has a lower nutrient content compared to the other organic fertilizers in this study. The macronutrient content, mainly N and K, is significant and present in higher concentrations than in the 20% nutrient solution, although nutrient values may vary considerably for reuse water depending on environmental characteristics and the treatment the effluent receives. The values found in this study were similar to those obtained by [81, 82], who used conventional treatments, and [83] whose treatment also occurred with the use of microalgae.

One of the major concerns regarding the application of fertilizers from the treatment of sanitary sewage is the potential presence of contaminants such as pathogens and heavy metals. Different studies demonstrate the efficiency of microalgae in disinfecting the environment by varying the conditions such as pH and oxygen concentration, and by intra-algal competition for nutrients and light [25, 84, 85]. Moreover, exposure to the sunlight and the drying process at temperatures above 55°C inactivate bacteria such as *Escherichia coli* and disinfect the effluent [86–88], thus ensuring the safety of products derived from sewage treatment for use in agriculture. The sludge generated from domestic sewage tends to have concentrations below the safety standards established by CONAMA [89, 90].

Plant growth is closely related to the availability of nutrients. The amaranth genus is known for its fast growth, with an accelerated growth phase occurring 30 days after emergence, and reaching physiological maturity 90 days after emergence [46]. In general, plants that received organic fertilizers together with 20% LA solution showed greater growth throughout the experiment, resulting in higher dry mass production. In addition to any mechanism of interaction between the different nutrients or of existing absorption, the accelerated growth may also indicate nutrient depletion from the organic fertilizers, which may not have been sufficient to meet the plant's nutrient requirements during the entire growth cycle, leading to nutrient deficiencies. Although the mineral content of the reclaimed water was lower than that of the 20% LA control solution, the plants that received only the former produced approximately 27% of the dry mass generated by the latter. This suggests that reclaimed water has potential as a biofertilizer Similarly, plants that received only sludge showed the potential to produce dry mass with the nutrients available in the sludge. Our study aligns with other studies that demonstrated the efficiency of using microalgae, biosolid and reclaimed water as biofertilizers for the growth of various crops [28, 59, 91, 92].

The low production of dry mass under pure sludge treatment may attributed to the low supply of K by the sludge, as the plants did not receive 20% LA. K plays an essential role in maintaining water status and the translocation and transformation of sugars, which directly impacts plant growth and development [93, 94]. On the other hand, the low production of dry mass in the treatment that exclusively received the reclaimed water, is mainly due to the deficiency of micronutrients, since Cu, Fe, Mn and Zn are below the concentrations of the 20% LA control, while the other nutrients reached the concentrations of the control treatment. As micronutrients play a crucial role in plant growth, development and metabolism, the deficiency of these essential nutrients negatively impacts dry mass production [95].

High levels of chlorophyll are essential for normal photosynthesis in plants. The adequate supply of nutrients, particularly N and Mg, is necessary for the synthesis of chlorophyll, as they are part of the molecule. Plants grown under wet microalgae showed a significant reduction in chlorophyll content even with adequate foliar N and Mg content, indicating that a large portion of these nutrients did not contribute to chlorophyll synthesis. When compared to plants grown under dry microalgae, those grown under wet microalgae had much lower foliar Fe content. Chloroplasts have the highest requirement of Fe in plant cells, comprising up to 80% of the cellular Fe in leaf cells [96]. Fe is essential for the biosynthesis of chlorophyll and its precursors and for the maintenance of the structure and function of chloroplasts [97–99]. Hence,

the lower concentrations of Fe in the plants grown under wet microalgae could have been responsible for the reduction in chlorophyll content. The lower photosynthetic pigments in this group of plants may be a consequence of rapid initial growth of plants, leading to rapid exhaustion of available nutrients in the wet microalgae with time. Although excessive Cu and Zn concentrations can significantly reduce chlorophyll content in plants [98, 100–103], this was not observed in plants grown under dry microalgae or sludge, which contained higher concentrations of Cu and Zn. The sludge had four times more Fe than in the dry microalgae in its composition, resulting in a higher content of Fe and chlorophyll in the leaves, highlighting the role of Fe in the chlorophyll biosynthesis. Photosynthetic pigments were consistently higher in crops watered with the treated wastewater [92]. In our study, plants cultivated with reclaimed water, chlorophyll content was higher than than those watered with 20% LA, possibly due to the increase in N and Mg levels, which are part of the chlorophyll molecule. Additionally, the observed decrease of carotenoids content in the leaves of plants cultivated under wet microalgae can be attributed to the role of Fe in carotenoid biosynthesis [98].

Photosynthesis can be limited by stomatal and non-stomatal factors. If $CO_2$ conductance is the only factor, limiting photosynthesis, $C_i$ would be expected to decrease due to $CO_2$ removal from intercellular spaces and decreased resupply through stomata. However, this was not observed in the plants grown under wet microalgae, where $C_i$ was increased or maintained at the same values as 20% LA control. This suggests that the reduction in photosynthesis was not solely due to stomatal closure but also due to lower levels of photosynthetic pigments and other non-stomatal limitations. Interestingly, despite the lower photosynthetic rate observed after 43 days of sowing, plants grown under wet microalgae showed an increase in shoot dry matter at the end of the experimental period. The effects of metals on photosynthetic capacity depends on their concentration availability. Lower concentrations of Zn have been shown to increase photosynthetic rate, $g_s$, $E$ and chlorophyll levels, while higher concentrations decreased them [104, 105]. Copper is an essential micronutrient for various metabolic processes in plants, but high concentrations can have a toxic effect on photosynthetic and biochemical parameters [106]. In our study, however, high doses of Cu and Zn from dry microalgae or sludge did not decrease photosynthesis. On the contrary, photosynthesis increased as the doses of dry microalgae and sludge increased. Our findings are consistent with previous reports that application of sewage sludge can increase photosynthesis, chlorophyll content, and Fe acquisition [53]. While plants grown under sludge supply showed lower $g_s$, photosynthesis increased, resulting in lower $C_i$, indicating effective $CO_2$ assimilation in the substomatal cavity. Plants grown solely with reclaimed water showed a 80% photosynthetic rate compared to the 20% LA control. However, the $g_s$ was lower, and $C_i$ reached a value similar to the 20% LA control, suggesting that the non-stomatal limitation was due to micronutrients deficiency in the reclaimed water.

## Conclusion

This study investigated the potential of organic fertilizers from domestic sewage treatment, such as biosolids, wet and dry microalgae and the reclaimed water to enhance the biomass of aerial parts of amaranth plants. The characterization of the organic fertilizer used in this experiment shows that their nutrient content is unbalanced in relation to plant needs. Despite the nutrient deficiencies observed in some treatments, the application of larger doses of these products significantly increased the aerial biomass production, with dry microalgae being the most effective. The partitioning of dry matter in the leaves was greater than in the stem probably due to N content in the biofertilizers used in this study. While the treatments receiving exclusively reclaimed water or sludge did not surpass the 20% control treatment, they

demonstrated the potential for biomass production with the available nutrients. Amendments with dry microalgae and biosolid increased photosynthetic rates with the biosolid being the most effective due to the increase in total chlorophyll content. To optimize initial growth, further investigations are needed to determine the appropriate doses and application rates of these products. Additionally, the high concentrations of micronutrients such as Cu, Fe, Zn, and Na in the biosolids and microalgae biomass require careful consideration to avoid potential stress on plants. Overall, our findings suggest that organic fertilizers from domestic sewage treatment can offer a sustainable solution for nutrient cycling by returning them to the soil and promoting plant growth in nutrient-poor soils.

## Supporting information

**S1 Table. Physical-Chemical characteristic of digested sewage.**
(PDF)

**S2 Table. Heavy metal concentration of digested sewage.**
(PDF)

## Acknowledgments

The authors thank the Department of Water and Sewage/Bauru for allowing us to carry out the research. The authors thank TANAC for providing the tannin (TANFLOC-SG).

## Author Contributions

**Conceptualization:** Inês Cechin, Gustavo Henrique Ribeiro da Silva.

**Formal analysis:** Elisa Teófilo Ferreira, Inês Cechin.

**Funding acquisition:** Gustavo Henrique Ribeiro da Silva.

**Methodology:** Sarah Corrêa Barrochelo, Sarah de Paula de Melo, Thainá Araujo, Augusto Cesar Coelho Xavier.

**Project administration:** Inês Cechin.

**Supervision:** Inês Cechin, Gustavo Henrique Ribeiro da Silva.

**Validation:** Elisa Teófilo Ferreira, Inês Cechin, Gustavo Henrique Ribeiro da Silva.

**Writing – original draft:** Inês Cechin.

**Writing – review & editing:** Elisa Teófilo Ferreira, Inês Cechin, Gustavo Henrique Ribeiro da Silva.

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
