## [Decision Letter · Decision Letter 0]

14 Jul 2023

PONE-D-23-17844Biofertilizers from wastewater treatment as a potential source of mineral nutrients for growth of amaranth plantsPLOS ONE

Dear Dr. Cechin,

Thank you for submitting your manuscript to PLOS ONE. After careful consideration, we feel that it has merit but does not fully meet PLOS ONE’s publication criteria as it currently stands. Therefore, we invite you to submit a revised version of the manuscript that addresses the points raised during the review process.

We look forward to receiving your revised manuscript.

Kind regards,

Sajid Ali

Academic Editor

PLOS ONE

Journal Requirements:

   "Author GHRS thanks the Brazilian agency FAPESP for financial support of the main project (Grant number: 18/18367-1), and the authors ETF and SCB thank for the undergraduate scholarships (Grant numbers: 2020/06459-9, 2020/10764-1, respectively).The author GHRS also thanks the Brazilian agency CNPq for financial support of the main project (Grant numbers: 308663/2021-7, 309064/2018-0, 427936/2018-7) and SPM thanks for the undergraduate scholarship (Grant number: 334). The authors thank the Department of Water and Sewage/Bauru for allowing us to carry out the research. The authors thank TANAC for providing the tannin. "

   "The funders had no role in the study design."

Reviewers' comments:

Reviewer's Responses to Questions

**Comments to the Author**

1. Is the manuscript technically sound, and do the data support the conclusions?

Reviewer #1: Yes

Reviewer #2: Yes

Reviewer #3: Yes

2. Has the statistical analysis been performed appropriately and rigorously? 

Reviewer #1: No

Reviewer #2: Yes

Reviewer #3: Yes

3. Have the authors made all data underlying the findings in their manuscript fully available?

Reviewer #1: Yes

Reviewer #2: Yes

Reviewer #3: Yes

4. Is the manuscript presented in an intelligible fashion and written in standard English?

Reviewer #1: No

Reviewer #2: Yes

Reviewer #3: Yes

5. Review Comments to the Author

Reviewer #1: The Manuscript Number PONE-D-23-17844 entitled 'Biofertilizers from wastewater treatment as a potential source of mineral nutrients for growth of amaranth plants' highlighted the importance of organic fertilizers from domestic sewage in enhancing the growth of amaranth. Definitely these types of studies offer a sustainable solution for nutrient cycling by returning them to the soil and promoting plant growth in nutrient-poor soils. However, there are some suggestions for the improvement of the submitted article

Major Suggestions:

1. The research gap has not been highlighted in the need for project section. So it is suggested to add already work done on the similar aspects and what needed

2. The statistical analysis is not quite clear. There is need to mention how data has been statistically analyzed separate section.

Minor Suggestions:

The minor suggestions is mentioned in the attached file

Reviewer #2: The manucript titled Biofertilizers from wastewater treatment as a potential source of mineral nutrients for

growth of amaranth plants is very interesting. However, the following justifications are much needed for the mansucript:

1. Why amaranth plants? Did the authors try with any other plants??

2. Mention the physiochemical composition of the watewater?

3. Was heavy metal analysis done for the wastewater?

4. The authors need to justify the reason for utlization of wastewater and address the constraint with the safety regulation related to the usage of wastewater.

5. The abstract and conclusion need to be improvised as they are too generalised and no information on the key experimental findings of the research.

Reviewer #3: The manuscript evaluates the sewage-grown microalgae as potential biofertilizer. The concept and results are provided in meaningful manner which will help in the real-time implementation of proposed biofertilizer. Various aspects of biofertilizers had been evaluated by authors which is good thing, however few recommendations are given below for the improvement of manuscript, prior to its acceptance for PLOS One Journal.

• It would be interesting to present the quantitative biochemical composition of Amaranthus in the introduction section.

• What was the reason of blocking light radiations during algal growth? Generally, higher light is associated with better nutrient removal and biomass growth.

• Please provide the details of the algal strain which is used in this experiment.

• Usually, algal growth is compromised in the presence of sludge or suspended solids. What was the behavior in present study? Which technique was used to obtain algal biomass in murky wastewater?

• Do the authors perform the safety assessment to evaluate the impact of tannin on biofertilizer, as this information is of importance for real-time application of proposed biofertilizer.

• Authors mentioned that 70% LA promotes the plant growth, then why this condition was not selected for the biofertilizer applications?

• Please provide the full form of abbreviation at the first mention of term.

• Why authors didn’t check the combination of reclaimed water and algae, as It would be interesting to evaluate their impact.

6. PLOS authors have the option to publish the peer review history of their article (what does this mean?). If published, this will include your full peer review and any attached files.

Reviewer #1: No

Reviewer #2: **Yes: **Dr. Archana Tiwari

Reviewer #3: No

---

## [Author Response · Author response to Decision Letter 0]

17 Oct 2023

Please, refer to the Response to Reviwers file.

---

## [Decision Letter · Decision Letter 1]

23 Nov 2023

Biofertilizers from wastewater treatment as a potential source of mineral nutrients for growth of amaranth plants

PONE-D-23-17844R1

Dear Dr. Cechin,

We’re pleased to inform you that your manuscript has been judged scientifically suitable for publication and will be formally accepted for publication once it meets all outstanding technical requirements.

Kind regards,

Sajid Ali

Academic Editor

PLOS ONE

Additional Editor Comments (optional):

Reviewers' comments:

Reviewer's Responses to Questions

**Comments to the Author**

1. If the authors have adequately addressed your comments raised in a previous round of review and you feel that this manuscript is now acceptable for publication, you may indicate that here to bypass the “Comments to the Author” section, enter your conflict of interest statement in the “Confidential to Editor” section, and submit your "Accept" recommendation.

Reviewer #2: All comments have been addressed

Reviewer #3: All comments have been addressed

2. Is the manuscript technically sound, and do the data support the conclusions?

Reviewer #2: Yes

Reviewer #3: Yes

3. Has the statistical analysis been performed appropriately and rigorously? 

Reviewer #2: Yes

Reviewer #3: Yes

4. Have the authors made all data underlying the findings in their manuscript fully available?

Reviewer #2: Yes

Reviewer #3: Yes

5. Is the manuscript presented in an intelligible fashion and written in standard English?

Reviewer #2: Yes

Reviewer #3: Yes

6. Review Comments to the Author

Reviewer #2: (No Response)

Reviewer #3: The authors are being appreciated for the subsequent revision of manuscript “PONE-D-23-17844_R1” according to the provided comments. It is now acceptable for publication in PLOS after some minor changes.

i. It is suggested to revise the 2nd sentence of abstract to “This study investigated the potential of sewage-derived biofertilizers on the growth and physiology of Amaranthus cruentus plants” for better syntax formation.

ii. Abstract should be in past sentence, please check and correct accordingly.

iii. 2nd line of introduction has to be checked again to confirm the intended meaning,

iv. Add just 1-2 lines at the end of 2nd paragraph of introduction to discuss the problems of conventional sewage treatment options that impels the research towards sustainable option of algal-based sewage treatment. It’ll help in establishing smooth linkage between various aspects.

7. PLOS authors have the option to publish the peer review history of their article (what does this mean?). If published, this will include your full peer review and any attached files.

Reviewer #2: **Yes**

Reviewer #3: No

---

## [Editor Report · Acceptance letter]

7 Dec 2023

PONE-D-23-17844R1 

Biofertilizers from wastewater treatment as a potential source of mineral nutrients for growth of amaranth plants 

Dear Dr. Cechin:

I'm pleased to inform you that your manuscript has been deemed suitable for publication in PLOS ONE. Congratulations! Your manuscript is now with our production department. 

Kind regards, 

on behalf of

Dr. Sajid Ali 

Academic Editor

PLOS ONE